# Three-Dimensional Cultivation a Valuable Tool for Modelling Canine Mammary Gland Tumour Behaviour In Vitro

**DOI:** 10.3390/cells13080695

**Published:** 2024-04-17

**Authors:** Mykhailo Huniadi, Natália Nosálová, Viera Almášiová, Ľubica Horňáková, Alexandra Valenčáková, Nikola Hudáková, Dasa Cizkova

**Affiliations:** 1Small Animal Clinic, University of Veterinary Medicine and Pharmacy, Komenskeho 73, 041 81 Kosice, Slovakia; mykhailo.huniadi@uvlf.sk (M.H.); natalia.nosalova@uvlf.sk (N.N.); lubica.hornakova@uvlf.sk (Ľ.H.); a.valencakova@uvlf.sk (A.V.); nikola.hudakova@student.uvlf.sk (N.H.); 2Department of Anatomy, Histology and Physiology, University of Veterinary Medicine and Pharmacy, Komenskeho 73, 041 81 Kosice, Slovakia; viera.almasiova@uvlf.sk

**Keywords:** cell cultivation, 3D cultivation models, CMT

## Abstract

Cell cultivation has been one of the most popular methods in research for decades. Currently, scientists routinely use two-dimensional (2D) and three-dimensional (3D) cell cultures of commercially available cell lines and primary cultures to study cellular behaviour, responses to stimuli, and interactions with their environment in a controlled laboratory setting. In recent years, 3D cultivation has gained more attention in modern biomedical research, mainly due to its numerous advantages compared to 2D cultures. One of the main goals where 3D culture models are used is the investigation of tumour diseases, in both animals and humans. The ability to simulate the tumour microenvironment and design 3D masses allows us to monitor all the processes that take place in tumour tissue created not only from cell lines but directly from the patient’s tumour cells. One of the tumour types for which 3D culture methods are often used in research is the canine mammary gland tumour (CMT). The clinically similar profile of the CMT and breast tumours in humans makes the CMT a suitable model for studying the issue not only in animals but also in women.

## 1. Introduction

The occurrence of mammary gland tumours among different mammalian species varies considerably. Tumours of the mammary gland are quite rare in mares, ruminants, and sows, but in domesticated bitches, they occur very often. In women, it is the most diagnosed malignancy worldwide, representing the most frequent cause of death. A great similarity between the CMT and human breast cancers (HBCs) does exist regarding incidence, behaviour, and histological origin. However, it is notable that the prevalence of CMTs is even more than three times higher than that of HBCs. CMTs account for approximately 40–50% of all tumours in dogs, and about 45–50% of them are malignant [1,2,3].

The most common form of malignant CMTs is simple carcinoma, which is derived from the glandular epithelium of the mammary gland. Another form of malignant CMTs that consists of both epithelial and myoepithelial components is complex carcinoma. CMTs that originate from mesenchyme–fibrosarcomas and other sarcomas are quite rare in bitches. Contra-wise, there are very often mixed CMTs (carcinosarcomas), which originate from both mesenchymal and epithelial components. Approximately two-thirds of dogs have multiple tumours, which means that different types of tumours occur within one animal [2,4,5].

The most important risk factors that promote the development of CMTs are age, breed, genetic predisposition, diet, different hormones, and growth factors [2,6,7,8]. Bitches, similar to women with preceding benign or pre-malignant lesions, seem to have an increased risk of developing new mammary cancer [9,10]. CMTs develop mainly in old or middle-aged bitches, with a median age of 9.5 years [10]. Among the cases of CMT were mainly purebred dogs, specifically Cocker Spaniels, German Shepherds, Yorkshire Terriers, Poodles, and Maltese [2]. The proliferative effect of ovarian steroids on the mammary gland epithelium may create conditions for carcinogenesis. In bitches, the majority of all mammary tumours are located in the posterior mammary glands (probably due to the higher volume of glandular tissue), and they appear shortly after the oestrus. While oestrogens mainly induce ductal neoplasia, progestins induce lobulo-alveolar neoplasia [11]. It is known that the serum levels of steroid hormones are higher in bitches with CMTs compared with healthy animals [12], and also the exogenous administration of progesterone derivates (used to prevent oestrus in bitches) may increase the risk of CMTs [10,13]. Obesity and a high-fat diet that causes the increased local production of oestrogen and leptin were evaluated as risk factors for the development of mammary gland tumour in women [14]. It can be assumed that obesity in dogs will have a similar effect on CMT development.

Malignant CMTs clinically manifest as single or multiple nodules in the glandular parenchyma, with or without the nipple’s affection. Rapid growth and ulceration are often typical features of malignancy. They can be inflamed with diffuse swelling, pain, and increased temperature. Metastases usually develop in lymphatic or blood vessels.

Diagnosis is based on clinical, blood, and serological examinations. A radiographic examination can evaluate the presence of metastases in the lungs and the increase in the size of lymphatic nodes. Prognosis and therapy are established according to anamnesis, clinical signs, and histological type of the lesion. In principle, CMTs can be evaluated by the speed of evolution, growth pattern, size, and presence of metastases. Differential diagnosis between benign and malignant CMTs is based on the immunohistochemical method (confirmation of collagen type IV expression for the identification of continuity of basement membranes). Morphometric methods, such as the evaluation of the numbers of nuclei per area or the evaluation of the minimal distance of cells from the basement membrane, are also used in CMTs [3].

Surgical excision of the tumour or mastectomy remains the most effective therapeutic method in most types of CMT, which also allows histopathological evaluation. Dogs with locally advanced diseases, aggressive types of tumours, or metastases need a more extensive therapeutic approach such as radiotherapy, chemotherapy, hormonal therapy, and others [15,16].

## 2. Cell Cultures for Cancer Research

Cell cultivation has become a fundamental technology in scientific research. Cell cultures are used to better understand the principal mechanisms underlying cell behaviour in vivo. They help reveal the physical and molecular mechanisms by which cells assemble into tissues and organs, how these tissues function, and how their function and structure change under the influence of various stimuli.

Currently, experimental studies are carried out either with the use of cell cultures collected directly from donors or established cell lines that are available in bioresource centres. Such cell lines are well characterised and are commonly used in research [17]. Isolation and cultivation of primary cultures obtained from living organisms/donors are usually more difficult because the tissue contains different cell subpopulations [18]. Cell culture management varies greatly depending on the type of cells chosen for the research [19].

Nowadays, cell cultivation is mostly used for cancer research, drug discovery, and stem cell studies. Since the 1900s, the 2D in vitro system has remained the most commonly used [20,21]. Due to the many limitations and the inability of 2D models to imitate the in vitro environment, research has shifted to the use of 3D systems, with the help of which we can better and more accurately replicate the natural environment in which cells live [22,23,24,25].

## 3. Two-Dimensional Culture, Pros and Cons

The 2D model system represents a monolayer of cells that are cultured either in Petri dishes or culture flasks and can adhere to a glass or plastic surface (Figure 1) [26,27,28,29]. The 2D model of cell cultivation has its advantages and disadvantages. It is a relatively cheap method with simple cell maintenance. In most cases, the cells are well characterised and easy to manipulate. On the other hand, 2D cultures are not able to imitate the real structure of the tumour or tissue. In this case, intercellular interactions are not the same as in the living organism, which affects important processes such as cell division, growth, differentiation, gene expression, and cell death [27,30,31]. Growth in a monolayer creates unlimited access to oxygen and nutrients, which does not represent the real in vivo situation due to the architecture of the tissue/tumour [31,32]. Two-dimensional culture systems represent adherent cultures, which are usually monocultures, and enable the study of only one type of cell. However, tumours have a complex or comprehensive structure, as they create a microenvironment that includes different types of cells. The tumour microenvironment influences the basic processes that take place in the pathological tissue and its survival. Therefore, it is crucial to understand the role of the tumour microenvironment in the pathogenesis of tumours [33,34].

Due to the disadvantages of 2D cultures, it was necessary to develop strategies to better understand the processes taking place in tumours, such as 3D cultures. Regardless of the existing shortcomings, 2D cultures are still used in science and research. Two-dimensional cultures are widely used in the study of cancer diseases, the development of new drugs, and personalised medicine. Moreover, 2D cultures are suitable for nonsolid cancer research, where we can include leukaemias, lymphomas, etc. [35,36]. Primary cultures obtained from thyroid tumours are used in personalised medicine and also in the testing of new antitumour agents [37]. Isolated peripheral blood mononuclear cells cultured in vitro can be used to monitor the response to immunotherapy or assess cell-mediated immune responses [38]. New substances that can have antitumour effects are also being tested on a 2D culture system. Such new molecules are either of natural origin or synthetically created. Examples can be chalcones and pyrrolidines, which are known for inducing apoptosis in tumour cells [39,40].

## 4. Three-Dimensional Models for Cancer Study

The development of 3D cultures has created a bridge between cell cultivation and in vivo studies [41,42]. In 1910, Harrison was the first to use the hanging drop technique, which he adapted from bacteriology to culture neurons in vitro [21]. Hamburger and Salmon were also among the first scientists to investigate the development of 3D cultures. In their work, they tested a medium based on soft agar, which supports the growth of human tumour stem cells. Such a cultivation method helped to better characterise different types of human tumours and is suitable for studies of new antitumour drugs [43]. Three-dimensional cultures should meet all criteria that are not achievable when using two-dimensional cultures, which means mimicking a specific tissue microenvironment or pathological microenvironment, where cells can grow, aggregate, and differentiate (Figure 2) [44]. The types of 3D cultures are summarised in Figure 3.

### 4.1. Scaffold-Free 3D Systems

Scaffold-free 3D systems represent a type of cultivation in specialised plates to which cells are unable to adhere and are forced to aggregate. This includes hanging drop microplates, low adhesion plates supporting the formation of spheroids, and others. Cultivation of cells in the form of spheroids makes it possible to restore the physiological microenvironment in which in vivo cell–cell interactions as well as interactions with the extracellular matrix are formed. The advantage of spheroids is that their size can be regulated by changing the density of the cell plating, which is related to the formation of nutrient and oxygen gradients, similar to living tissues [45,46,47,48,49].

Hanging drop is a method of cultivation in which cells aggregate without an attachment surface. For this type of culturing, specialised bottomless plates are used, which enable the growth of cells in a small drop of media. The resulting spheroids thus consist of several layers of cells. There is the outer layer with active proliferating cells as a result of easily accessible nutrients and oxygen, the middle layer of quiescent cells, and a necrotic core [45] (Figure 2). Hanging drop cultures are used in tumour research, cardiac spheroid engineering, and hepatotoxicity testing [50,51,52].

Magnetic levitation represents the formation of spheroids from cells, which are filled with magnetic nanoparticles. Subsequently, by applying an external magnetic field, the cells are levitated towards the air/liquid interface within the plate with low adhesion [53]. This promotes the formation of cell-to-cell contacts and supports the formation of spheroids. Magnetic levitation can be used in breast cancer research. Cells are incubated with nanoparticles/nanoshuttles and then a magnet can be used to connect the cells and create a 3D structure. Moreover, chimeric 3D tumour mass can be created by adding different cell types, fibroblasts, for example [54,55]. Recent studies have found that magnetic levitation can also be used in the detection of cancerous diseases in the early stages and help in establishing a diagnosis [56,57].

Originally, an organoid can be defined as a 3D system that is made of pluripotent tissue-specific cells that perform at least one function of the organ from which they originate. The disadvantage of organoids is that they are dependent on growth factors, the extracellular matrix (ECM), intercellular interactions, and the ability to model immune responses [58,59,60]. Organoids can be developed from various tissues, such as the brain, liver, intestinal, retinal, mammary, etc. [61,62,63,64,65]. Such diversity of organoids allows their wide use in toxicity research, the development of new drugs, and personalised and regenerative medicine [66].

### 4.2. Scaffold-Based 3D Systems

#### 4.2.1. Biological 3D Scaffolds

Scaffold-based systems provide important physical support for cell cultures. Scaffolds can be of various origins, i.e., biological or synthetic, designed according to the required properties of the ECM (charge, adhesiveness, stiffness), which is necessary for cell aggregation, growth, and migration [67,68,69]. Some scaffolds may contain hormones, growth factors, or other biologically active substances that influence cell characteristics, proliferation, gene expression, and ultimately their specific phenotype [68]. These models are used for creating co-cultures of malignant and normal cells as they mimic the heterogeneous structure of a tumour, understanding the role of stromal cells during tumourigenesis. Therefore, recent studies based on scaffolds were focused mainly on the cultivation of primary tumour cells, testing of new drugs on donor samples, and personalised medicine [70,71,72].

Hydrogels of biological origin represent hydrophilic networks connected by covalent bonds. Natural hydrogels are characterised by good biocompatibility and are easily adjustable from a biophysical and biochemical point of view [73]. In practice, combined hydrogels containing components of biological and synthetic origin are also used, thereby gaining new advantages. Natural hydrogels of animal origin include scaffolds based on collagen, fibrin, chitin, gelatine, plants such as alginate, and Matrigel, which is enriched in laminin, collagen type IV, perlecan, and entactin. Collagen, fibrin, and Matrigel are characterised by the ability to maintain and support cell functions (viability, growth, controlled differentiation) [45,74]. A typical natural hydrogel and at the same time the best characterised is hyaluronic acid. By modifying glycosaminoglycan with various functional groups, we can obtain peptide hydrogels with completely new characteristics, usable for various applications. Jakubikova et al. pointed out the resistance of multiple myeloma cells to chemotherapeutic drugs, which were co-cultivated in a 3D hydrogel-based culture of mesenchymal stem cells (MSCs). Drug/chemotherapeutic resistance of primary cultures of multiple myeloma makes it possible to study this process in vitro, test new drugs, and provide information for personalised medicine [75].

Collagen I-based scaffolds are the most common and widely used for tissue engineering such as bone bioengineering, tissue engineering for musculoskeletal deficiencies, and demineralised bone powders and porous collagen devices [76,77,78]. Matrigel represents a permeable 3D system enabling the transfer of nutrients through its network and thus supporting cell growth. Like collagen hydrogels, Matrigel serves as an ECM replacement and is used in disease research. The attachment of cells using Matrigel and collagen is facilitated by integrin receptors, which results in the activation of signalling pathways affecting basic physiological processes in cells and also modulating the response of cells to chemotherapy, radiation, and immunotherapy [79,80].

Fibrin is a natural polymer obtained via polymerisation of the plasma protein fibrinogen in the coagulation process. Fibrin is most often used in studies of angiogenesis, MSCs, and biomechanical studies. Due to the high degree of degradation by proteases, its use is limited in long-term cultivations [81,82,83,84]. In combination with polyethylene glycol, fibrin was used in a breast cancer study. The combined scaffold increased the viability of breast tumour cell lines (MCF-7, MDA-MB-231, and SK-BR-3) depending on the size of the colony and their morphology [85]. There are other scaffolds of animal origin, such as gelatine, which is created in the process of hydrolysing collagen protein gelatine, or chitin, which due to its structure can increase cell proliferation, is nontoxic, supports regeneration, and has antibacterial effects [86,87].

Alginate is a polysaccharide extracted from brown algae. It is well known for its nontoxic properties, good biodegradability, and is composed of mannuronic and guluronic acids. In practice, its use in 3D cultures is limited due to rapid degradation. On the other hand, it is often used in 3D bioprinting [88,89], protein, nucleic acid, and cancer drug delivery systems [90,91,92]. In addition, alginate hydrogels can be used in the treatment of tumour diseases using microwave ablation to increase its effectiveness and minimise tumour recurrence after treatment [93].

The type of matrix used in 3D cell cultures is often determined by the type of cells being cultured. In vivo, different types of cells interact differently with their surrounding ECM; therefore, simulating this interaction in vitro is crucial for maintaining cell viability, function, and phenotype. For instance, collagen is a major component of the ECM in many tissues, including bone, cartilage, skin, and blood vessels and is commonly used as a matrix for the 3D culture of cells such as fibroblasts, chondrocytes, endothelial cells, and keratinocytes [94]. Matrigel is predominantly used for culturing epithelial cells, endothelial cells, and stem cells [95]. On the other hand, alginate hydrogels are biocompatible and have flexible mechanical properties, making them suitable for encapsulating various cell types, including stem cells and pancreatic islet cells [96]. Lastly, fibrin hydrogels are commonly used for culturing endothelial cells, smooth muscle cells, and cardiomyocytes [97]. 

Overall, the choice of matrix for 3D cell cultures depends on several factors including the desired cell phenotype, tissue-specific ECM composition, mechanical properties, and experimental objectives. Whether it is for tissue engineering, drug delivery, or cell culture, the intended application will influence the choice of hydrogel material. The mechanical properties required for intended application should be considered. Characteristics like the mechanical strength, degradation rate, and the evaluation of hydrogen and cell interaction are important factors to acknowledge when choosing a matrix [98].

#### 4.2.2. Synthetic 3D Scaffolds

As already mentioned, the ECM plays an important role in supporting basic physiological processes in cells and is very heterogeneous. Therefore, many synthetic scaffolds have been developed in an attempt to mimic the complex structure of the ECM in vitro. Overall, all synthetic scaffolds can be divided into two groups: natural and artificial synthetic polymers. Among the most used synthetic polymers are polyglycolic acid (PGA), polylactic acid (PA), and polyethylene glycol (PEG). The advantage of these substances is that they are cheap, easy to manipulate, and are also characterised by good reproducibility, thus promoting consistent results [99,100,101].

#### 4.2.3. Decellularised Matrices

Decellularisation scaffolds are created by eliminating genetic material and native cells from the ECM, while the key is to preserve their biomechanical, biochemical, and structural properties. The resulting cell-free carcasses can be filled with cells directly from the patient for use in personalised medicine [102].

Decellularised scaffolds (DSs) can be of animal or human origin. In practice, these DC-based 3D models are seeded with the patient’s cells for cell culturing to create personalised autologous tissue or organ transplantation therapy. Also, recellularised matrices from various sources can be applied as 3D ex vivo models for disease research, hydrogel synthesis, tissue engineering, and 3D printing [103,104,105]. The latest reports describe the use of a decellularised matrix in metastatic formation research, chemotherapy treatment response, and ECM role in tumour development. For example, 3D organoids cultivated in decellularised scaffolds are usable for colorectal cancer research, reconstruction of cervical cancer tissue, and breast cancer research [106,107,108].

### 4.3. Specialised 3D Cell Culture Platforms

#### 4.3.1. Microfluidic Devices

Compared to other 3D culture systems, microfluidic devices enable a continuous supply of oxygen and nutrients for the cells and at the same time remove waste materials that are produced during biological processes. The continuous application of drugs or other biologically active substances allows the use of microfluidic devices in the study of new potential therapies, cancer research, tumouroid cultures, and screening of small molecules [45,109,110,111].

The ability of microfluidics, a technology that manipulates and regulates fluids at the micron scale, to produce homogenous microspheres with precise geometry has made it a viable technique for creating hydrogel microspheres. Based on the methods of production, there are three types of microfluidics that are currently available, namely, additive manufacturing, modular assembly, and micro-processing. The development of capillary-based microfluidics makes it possible to produce micro-materials in regular laboratory settings, which significantly advances the application and acceptance of microfluidic technology [112].

One of the advantages of microfluidic devices is the possibility of designing them according to the requirements of the experiment. In general, a microfluidic device should consist of microchambers and microchannels imitating the structure of an organ/tissue. Another advantage is the small volume of chemicals used when monitoring the basic pharmacokinetic and pharmacodynamic parameters of newly tested drugs [113,114,115]. Yang et al., 2015, studied the efficacy of photodynamic therapy (PDT) on breast tumours using the co-cultivation of breast tumour cells together with adipose tissue-derived stem cells in a microfluidic system. The principle of PDT is the application of photosensitive substances that create active forms of oxygen in the target tissue after irradiation. Moreover, compared to conventional chemo- and radiotherapy, PDT has significantly fewer side effects. After applying PDT to a microfluidic system with co-cultured breast tumour cells and stem cells, the tumour cells had a higher survival rate in the 3D system compared to the data contained in the study on the 2D cultures [116]. 

Frequently, microfluidic systems are used to study the effectiveness of nanoparticles in tumour research. Modification of nanoparticles like the targeting of cells/specific molecules using their binding to specific sites, or the release of drugs that are bound to nanoparticles, enables their use for various purposes [117,118]. An example can be the conjugate of 6-mercaptopurine and carboxymethyl chitosan, which is used for leukaemia therapy. A study showed that the drug release was increased in tumour cells compared to normal cells [119]. In view of these findings, microfluidic devices can be used not only for testing new drugs, but also for the creation of new techniques for the early diagnosis of tumour diseases, detection of circulating tumour cells, and changes in the expression of tumour biomarkers.

#### 4.3.2. Organ-on-Chip

In recent years, Organ-on-chip (OoC) technology has received more attention, mainly due to stem cell availability and the establishment of international OoC programmes. Standardisation of procedures, independent device testing, and qualification are essential for reaching the full potential of this 3D model system in disease modelling, drug development, and personalised medicine [120]. The main goal of OoC is to mimic aspects of human pathology and physiology, which will provide better model systems for science and research. Existing models such as cancer-on-chip, vessels-on-chip, neurons, and glial cells-on-chip, lung-on-chip, and ALS-on-chip [121] are novel technologies that are already widely used in research and industry. An increasing number of OoCs are based on induced pluripotent stem cells (iPSCs), cell lines, primary cultures, and organoids; therefore, the models vary from single-organ systems to body-on-chip [122,123,124,125]. Importantly, human cell-based OoCs would also reduce the exploitation of animals in research [126]. Commonly, OoC technology uses spheroids that have been developed and cultured on a chip to mimic the tumour microenvironment. This can be achieved by co-cultivation with other cell types, typical of the tumour microenvironment, by vascularisation, or by culturing primary cells isolated from the tumour. Moreover, new technologies that use the OoC method are constantly appearing [127,128].

While the basic OoC can imitate only one organ, multi-organ chips allow the adaptation of each compartment of the 3D model to a different organ. It is useful when testing new drugs, where one part of the chip is responsible for its absorption (gut), the second for metabolism (liver), and the third for elimination (kidneys) [129]. Maschmeier et al. created a four-organ chip for the co-cultivation of epithelial, intestinal, liver, and kidney cells. With the help of genetic and metabolic analysis, it was found that this system ensured homeostasis in the co-culture of four tissues, which can be maintained for up to 28 days [130].

OoC is commonly used to study tumour metastasis and extravasation. Jeon et al. 2015, investigated different degrees of extravasation of breast tumour cells in tissue-specific scaffolds derived from bone and muscle. The extravasation process was significantly higher in the case of a microenvironment imitating bone tissue [131].

In the process of drug development, the OoC method holds its deserved place. The ability to mimic organ physiology makes it possible to include this technology in almost every key process of discovering new compounds, from early screening to preclinical trials [132]. Multi-organ chips are an excellent model for obtaining data on the pharmacokinetics and pharmacodynamics of new test substances in situ. The Shuler group were the first to report the use of a 3D tumour, liver, and bone marrow chip to study the toxic effects of 5-fluorouracil. The compartment representing the liver showed greater resistance than the bone marrow [133].

Over the past few years, OoC technology has integrated many new concepts and been used in practice. Currently, like other 3D cultures, it can be used at various levels for cancer research, especially for the development of new drugs. Therefore, OoC has a significant potential in the pharmaceutical industry as well as in personalised medicine.

#### 4.3.3. Three-Dimensional Bioprinting

The first 3D printer was invented by engineer Charles Hull in the 1980s, which was able to create solid objects based on computer-aided design (CAD) [134]. In the late 1990s, 3D printing began to be used in healthcare, most often in surgery in the production of prosthetics, dental implants, etc. Subsequently, the new term “bioprinting” was introduced, when living cells, active biomolecules, or other biomaterials, generally known as “bioink”, were used as material. The result of stacking such “bioink” layer by layer is the creation of 3D structures such as organs and tissues [135,136].

Three-dimensional bioprinting is widely used in cancer research and the pharmaceutical industry. A huge advantage of 3D bioprinting is the possibility of the controlled creation of cell structures (mix of tumour cells and biomolecules) into a predefined hierarchy [137]. For instance, HeLa cells can be used as a bioink for designing 3D structures in cervical cancer research. Zhao et al., 2014, used 3D printing to encapsulate HeLa cells in a hydrogel. Next, the 3D structures were compared with classic HeLa 2D cultures, and significant differences were observed, such as increased proliferation, metalloproteinase (MMP) protein expression, and the chemoresistance of the cells against paclitaxel [138]. Hong and Song report on the possibility of investigating the resistance of tumour cells to therapy using 3D bioprinting. In their study, they formed resistant spheroids derived from MCF-7 tumour stem cells (breast cancer cell line) using 3D bioprinting, and bulk MCF-7 spheroids, which were treated with paclitaxel and camptothecin (both agents have antitumour effects). During the experiment, it was found that the EC_50_ values of both substances were significantly higher for the 3D-printed spheroids than for the bulk tumour spheroids [139].

## 5. Two-Dimensional and Three-Dimensional Model Systems in CMT Research

As already mentioned, 2D and 3D model systems are widely used in the research of human tumour diseases. Three-dimensional cultivation helps to better understand the processes taking place in the microenvironment of human tumours, testing of new promising drugs, and personalised medicine. But despite this, today, we often encounter preclinical trials based on 2D cultivation. Canine tumour research is no exception. As time goes on, new articles about research on various types of canine tumours, such as CMT, osteosarcoma, testicular tumours, etc., accumulate [2,140,141,142]. Moreover, CMT can serve as a model for human breast cancer research in translational oncology. The reason is the clinically similar profile of CMT and breast cancer in women, which includes risk factors, histology, expression pattern of hormonal receptors, and genetic characteristics [143,144,145,146]. CMT develops much faster compared to breast cancer in women, which is an advantage in the study of this disease. On the other hand, researchers encounter various limiting factors. Bitches can develop several types of CMT tissue, which subsequently affect the results. Postoperative treatment is not always carried out in clinics, which limits the obtaining of homogeneous results; also, many young bitches undergo ovariohysterectomy, so this should be considered in comparative studies. And the important point is that dogs are not used as experimental animals for ethical reasons. Therefore, the isolation of CMT cells and the development of 3D in vitro models is an innovative method for the study of spontaneous breast cancer [147,148,149,150].

### Microenvironment/Cellular Components in 3D Culture

When creating a 3D cell culture microenvironment to mimic canine mammary gland tumours, several factors must be optimised to replicate the physiological conditions as closely as possible. It is necessary to consider components of the native tumour, including cell–cell interactions, extracellular matrix composition, gradients of oxygen, nutrients, and signalling molecules. This makes it possible to study tumour behaviour, angiogenesis, and immune response and at the same time the development of targeted therapeutic interventions. Cell lines from malignant mixed mammary tumours or metastatic mammary adenocarcinoma tumours such as CMT-U27, CMT-U309, and CHMp provide useful instruments to study the biology of malignancies, including invasion, metastasis, proliferation, and response to treatment [151,152,153].

Using primary cultures isolated directly from canine mammary tumours is more convenient and has several advantages, including the preservation of diverse cell types, genetic mutations, microenvironmental cues, and relevance to clinical samples. Furthermore, they better capture the complexity of tumours than cell lines that have the potential to experience clonal selection and adaptability over extended periods of culture. With the use of primary cultures, it is possible to generate more physiologically appropriate models for investigating tumour biology, drug screening, and personalised medicine techniques in canine mammary gland tumour research [154]. 

Stromal cells (fibroblasts, cancer-associated fibroblasts (CAFs), MSCs, endothelial cells, and pericytes) play a crucial role in supporting tumour growth and progression by facilitating the development of blood vessels and creating an environment conducive to tumour progression [155]. Numerous pro-angiogenic molecules, including vascular endothelial growth factor (VEGF), fibroblast growth factor (FGF), platelet-derived growth factor (PDGF), and angiopoietin-1 (ANG-1), are secreted by fibroblasts and infiltrated immune cells [156]. Therefore, co-culturing has been widely used in 3D cancer models to mimic stromal–cancer cell interactions. Cancer-associated fibroblasts influence every key process in the tumour such as proliferation, invasion, migration, and apoptosis via cell–cell communication [157]. Ma et al. cultured CAF cells together with gastric tumour cells using the hanging drop technique. CAF cells supported the growth of spheroids and their diameter increased significantly when compared to tumours lacking CAFs, after 72 h of cultivation [158]. Nasiraee et al. studied the invasiveness of MDA-MB cell tumouroids treated with all-trans retinoic acid (ATRA) and seeded on a CAF-derived hydrogel matrix in a microfluidic device. They showed significantly decreased invasiveness of MDA-MB spheroids after treatment with ATRA [159]. Most tissues contain MSCs, which primarily support healing processes and also form the milieu surrounding tumours.

Because of this, MSCs are widely used for 3D cultivation along with cancer cells, mainly in scaffolds and the hanging drop method [160]. However, previous studies indicate contradictory effects of MSCs on the tumour structure. In some cases, MSCs contributed to the inhibition of tumour progression through cell cannibalism; in others, they supported its survival, chemoresistance, and invasiveness [75,161,162].

Stromal cells also regulate the immune system’s reaction to malignancies. They can either increase immune surveillance to prevent tumour growth or reduce antitumour immune responses to promote the opposite effect. In this context, stromal cells can control the immune response in the tumour microenvironment, creating an immunosuppressive milieu that promotes angiogenesis and tumour growth. Furthermore, they release chemokines and cytokines that attract regulatory T cells and macrophages, two types of immune cells that secrete pro-angiogenic substances and stimulate angiogenesis. 

In vivo, macrophages are the most represented immune cells in the tumour microenvironment. Infiltration by macrophages promotes angiogenesis and the formation of metastases; therefore, it is associated with a poor prognosis and progression [163]. It has been shown that CD14-positive monocytes selectively supported the invasion of THP1-malignant epithelial non-polarised cells, but not normal polarised epithelial cells, in a Matrigel-based 3D system. Li et al., 2017, found that non-polarised breast tumour cells produced more reactive oxygen species (ROS), which promoted tumour progression, compared to polarised cell lines in 3D cultures, accompanied by the induction of NF-κβ factor and cytokine expression. Subsequently, it was shown that the loss of polarisation and increased ROS levels supported monocyte infiltration in the 3D co-cultivation process [164]. Such co-cultured 3D systems can be a promising model for studying the invasiveness of different types of solid tumours.

T lymphocytes in the tumour microenvironment can either have an antitumour activity or suppress the activation of CD8+ cytotoxic T lymphocytes. Activation and differentiation of infiltrated T lymphocytes in the tumour also depend on other stromal cells. Koech et al. observed the changes in the ability of T lymphocytes to infiltrate the tumour by co-cultivation of A549 and Calu-6 tumour cell lines with the SV80 fibroblast cell line. After tumouroid formation, peripheral blood mononuclear cells (PBMCs) were added and tumouroid infiltration was analysed immunohistochemically after 24 h. As a result, it was shown that fibroblasts supported tumouroid infiltration by cytotoxic T lymphocytes. Moreover, the tissue migration of CD8+ T lymphocytes was observed [165]. Immune cells can be cultivated in a 3D environment to investigate these intricate interactions and create immunotherapies. Stromal cells have the power to affect how cancers react to anticancer treatment. For instance, they may, through a variety of pathways, result in drug resistance or establish a protective milieu that shields tumour cells from treatment.

The cellular microenvironment has an important role in the processes that take place in the tumour, but the intercellular space and the matrix formed by acellular structures are no less important. In the modelling of 3D cultures, polymers of various origins are widely used to create such acellular structures into which cells are subsequently seeded. Although collagen gels, Matrigel, and other polymers are often used as ECM templates, many of them do not represent the true composition of the ECM of the tumour. Therefore, in 3D culture modelling, decellularised tissue is often used to mimic the in vivo ECM microenvironment of the tumour [166]. This technique was used for the first time by Livesey et al., 1995, to create a scaffold for skin regeneration. Later, this procedure was also applied to other organs such as the heart valve and bladder. From the standpoint of studying malignancies, these scaffolds are used in the research of invasiveness and metastasis [167,168,169].

By colonising the decellularised lung tissue, the invasive ability of the breast tumour can be studied ex vivo. It was shown that tumour cells formed metastases in an experimental microenvironment, and after silencing the Zeb1 protein, which is involved in the process of epithelial–mesenchymal transition. This ability was significantly reduced, which emphasises the importance of factors and proteins present in the ECM formed from the target tissue [170]. Similar results were also found in other types of tumours, such as colon and pancreatic cancer [171,172].

Cancer stem cells (CSCs) found in canine mammary gland tumours can proliferate and differentiate, which gives them the ability to initiate new tumours. These cells can develop from dedifferentiated, more developed tumour cells, or normal mammary stem cells. Because CSCs may regenerate the tumour hierarchy, it is believed that they are the cause of both tumourigenesis and recurrence. Tumour heterogeneity in canine mammary gland tumours is partly attributed to the existence of stem cells. The several cell populations they give rise to, each with distinct phenotypic and functional features, are the cause of intratumoural heterogeneity. This heterogeneity can affect tumour behaviour, treatment response, and disease progression [173].

To manage canine mammary gland cancers more effectively, it is imperative to comprehend the role that stem cells play in tumour biology. The microenvironmental cues present in canine mammary gland tumours encompass a complex network of cellular and molecular interactions within the tumour microenvironment. The ECM supplies biochemical cues and structural support that control the activity of tumour cells. Changes in the content and stiffness of the ECM impact the proliferation, migration, invasion, and metastasis of tumour cells in canine mammary gland tumours. Low oxygen tension, or hypoxia, is a characteristic that many solid tumours share, including mammary gland cancers in dogs. Angiogenesis, metabolic reprogramming, tumour development, and treatment resistance are all facilitated by hypoxic areas in the tumour microenvironment. Canine mammary gland tumour formation and progression are linked to chronic inflammation. Tumour growth, invasion, and metastasis are encouraged by the cytokines, chemokines, and growth factors produced by inflammatory cells such as neutrophils, lymphocytes, and macrophages. Numerous immune cell types, such as tumour-infiltrating lymphocytes (TILs), regulatory T cells, and myeloid-derived suppressor cells (MDSCs), are present in the immunological microenvironment of canine mammary gland tumours. Tumour and stromal cells interact with immune cells to influence immunotherapy efficacy, immune evasion, and antitumour immune responses. The hallmarks of canine mammary gland cancers include enhanced glycolysis, which is a modification in the tumour cell metabolism. Tumour growth and survival are enhanced by metabolic reprogramming, which enables tumour cells to adapt to the hypoxic and nutrient-deficient environment of the tumour [174].

In the tumour microenvironment, stromal cells and tumour cells release extracellular vesicles (EVs), which include exosomes and microvesicles. In canine mammary gland tumours, EVs transport bioactive substances, including proteins, nucleic acids, and lipids, which facilitate intercellular communication, encourage tumour–stromal interactions, and alter the course of tumour growth [175].

In a 3D culture system of canine mammary gland tumours, spatial gradients of oxygen, glucose, and other nutrients can be established to mimic the metabolic variability and hypoxic microenvironment of solid tumours [176]. When comparing the degree of hypoxia in 2D and 3D breast tumour cultures, it was shown that there is a higher degree of hypoxia in denser 3D tumouroids compared to 2D cultures. Hypoxia takes place in the centre of tumouroids, which imitates the process taking place in a solid tumour [141]. It is known that hypoxia is one of the factors associated with resistance to treatment; therefore, 3D modelling contributes to the development of more effective therapeutic methods by providing important insights into the biology, metabolism, and therapeutic response of malignancies. This methodology contributes to the development of more effective therapeutic methods by providing important insights into the biology, metabolism, and therapeutic response of malignancies.

ECM remodelling is also facilitated by stromal cells, which release proteolytic enzymes, such as tissue inhibitors of metalloproteinases (TIMPs) and MMPs. These enzymes help break down and reorganise the ECM to create space for the formation of new blood vessels [177,178].

## 6. Application of Different 3D Cultivation Models in CMT Research

Spheroids are simple cell clusters that can be formed from a broad range of tissues, typically collected via the biopsy of diseased tissue or tumour. Thus, they can be created from primary cell cultures obtained from donors and patients [179]. Cardoso et al. studied gene expression in primary lines of CMT, specifically simple carcinoma (SC) and complex carcinoma (CC). The expression was compared on cell cultures cultivated based on 2D models and 3D scaffolds, where spheroids of different sizes were created. Over 14 days, mRNA and protein concentrations were compared in both cell lines. Higher levels were detected in the SC 3D and 2D models compared to CC. In addition, higher concentrations of relaxin receptors 1 and 2, MMP -1,-2,-9, and -13 were observed in 3D models compared to 2D in all cell lines. The exception was E cadherin, whose expression was downregulated in 3D models, while epidermal growth factor receptor (EGFR) was expressed on SC and CC cells in both model systems [180]. EGFR plays an important role in breast cancer in humans and dogs, supporting pathological cell survival, growth, and proliferation [181,182]. Similarly, MMPs are responsible for the remodelling of connective tissue and the formation of metastases. Together with relaxin, whose expression is increased in human breast cancer, they represent independent markers for the formation of metastases in female dogs with CMT [183,184,185].

Unlike spheroids, organoids represent much more complex structures consisting of specific cells and ECM components in such a way as to imitate original tissue (stomach, skin, liver, etc.) [179]. Organoids derived from CMT cells are used in human breast cancer research. The advantage of these in vitro tools is the ability to imitate important immunohistological and morphological features of the tissue from which they were derived. In addition, in the case of CMTs, it is possible to obtain normal tissue or benign lesions from the same patient, which is a great advantage. The genetic characteristics of CMT organoids are preserved as in primary tissue. Among these, we can include PIK3CA carcinomas, which are positive for the estrogenic receptor. Practically, such carcinomas can be used as a model in humans for preclinical studies of breast cancer.

In addition, Inglebert et al., 2022, reported on the possibility of modifying CMT organoids using CRISPR/Cas9 technology [143]. Therefore, new drugs, the influence of genetic polymorphisms on therapeutic strategies, response to treatment, and prediction outcome can be tested on them [143]. Using a long-term culturing protocol for human mammary organoids can create a living biobank that captures the heterogeneity of the disease. Such organoids recapitulate the histopathology of the original tissue as well as prominent genetic factors, e.g., HER2 status and hormone receptor status [186]. It was found that even after the long-term passaging of HBC organoids, the genetic material remains in a consistent state and no significant changes occur in the DNA. Due to the preserved gene expression, the use of HBC organoids in in vitro drug screening, evaluation of response to treatment, and in personalised medicine, which is consistent with the patient response and in vivo xenotransplantation, is possible [187].

Three-dimensional cultures are widely used in evaluating the invasiveness of tumour cells and their ability to form metastases. A typical method is the passage of cells through a porous membrane, which is coated with a biomaterial resembling the ECM, for example, Matrigel [188]. A similar method was used by Manuali et al., 2012, when comparing the invasiveness of several CMT cell cultures. Cells were incubated under standard conditions and subsequently treated with calcein-AM, which is hydrolysed in the cytoplasm to a polar fluorescent product. To confirm the invasiveness of the tested cell cultures, a Matrigel matrix was used on which some tumour cultures formed colonies and some branching structures. This testifies to the invasive phenotype of these cancer cell lines [189]. Similar results were reported by Krol et al., 2012, during the cultivation of CMTs under normal conditions, with the exception that not all CMTs could migrate. After co-cultivation of the tested cancer lines with macrophages, a higher migration ability was observed. A Matrigel matrix was also used to confirm the results, where their invasive phenotype of CMT cells was confirmed [190].

Decellularised ECMs from rat or human mammary gland tissue can serve as a basis for the formation of hydrogels. Such matrices support the growth of not only the tumour but also normal epithelial breast cells. ECM hydrogels are a suitable scaffold for the cultivation of these cells due to the preserved signalling reactions taking place in the breast tissue. Mollica et al., 2019, describe the possibility of cultivating large tumouroids/organoids created using 3D [191] bioprinting in such tissue-specific bio scaffolds, which has great potential in the study of mammary gland tumours. ECM factors play a crucial role in cell differentiation and proliferation and are also tissue-specific, which is why understanding the role of the ECM in breast cancer is one of the main assignments of modern breast cancer research [192].

In the process of studying the tumourigenesis of breast cancer, a phenomenon was discovered in which the normal breast microenvironment can suppress the progression of this disease and convert the pathological phenotype of tumour cells to normal [193]. Therefore, understanding this phenomenon can be of great importance in the development of new detection and treatment strategies. In context with these findings, 3D bioprinting provides us with new platforms for studying the process of the microenvironmental switch of pathological breast cells. Reid et al., 2019, pointed out the possibility of creating and cultivating tumouroids using a 3D printer in collagen gels. Moreover, they demonstrate the production of chimeric tumouroids, which also contain normal breast epithelial cells. Chimeric structures had a significantly increased level of 5-hydroxymethylcytosine (takes place in gene transcription) compared to tumouroids, which shows that a normal microenvironment causes cancer cells to redirect to normal [194]. These results indicate the potential use of this technology in the study of the microenvironment of the breast tissue.

One of the biggest complications for patients with HBC is the formation of metastases [195]. OoC technology can provide information on the tissue-specific metastatic potential of HBC cells [196,197]. Firatligil-Yildirir et al., 2021, tested two OoC platforms, specifically extravasation (EX-chip) and invasion/chemotaxis (IC-Chip) to rate the extravasation and invasion of liver, lung, and breast tissue (extravasation is the process of tumour cells invading from the interior of a vessel into the organ parenchyma). In the simulated microenvironment of individual tissues embedded in Matrigel, the invasiveness and extravasation of two types of HBC were monitored: non-invasive MCF-7 and invasive MDA-MB-231. As a result, it was found that it is possible to distinguish the metastatic phenotypes of tumours using EX and IC chips that stimulate the tissue-specific microenvironment. These results can help in the diagnosis of the disease and the prediction of the occurrence of metastases, which will allow us to choose the most appropriate therapy for individual patients [197].

## 7. Conclusions

In science, tumour diseases have posed one of the most significant challenges in medical science for decades. Researchers are constantly working to gain a deeper understanding of the mechanisms underlying the initiation, progression, and metastasis of cancer, as well as to develop more effective prevention, detection, and treatment strategies. Nowadays, new tools for tumourigenesis research are emerging, including 3D cultures. With the advancement of this new technology, we can better understand the role of the ECM, the formation of metastases, test targeted therapies, and personalised medicine, which aim to specifically target cancer cells while minimising damage to healthy tissues. The possibility of 3D cultivation using primary cultures brings us much closer to animal models than 2D in vitro studies, which is why 3D models are widely used in tumour research in animals. Canine mammary tumours serve as an excellent model for studying certain aspects of HBC. Both CMTs and HBC share similarities in terms of their genetic markers and pathophysiology, making CMTs a valuable research tool in oncology. However, several obstacles persist, including treatment resistance, cancer heterogeneity, and the requirement for enhanced prognostic biomarkers.

## Figures and Tables

**Figure 1 cells-13-00695-f001:**
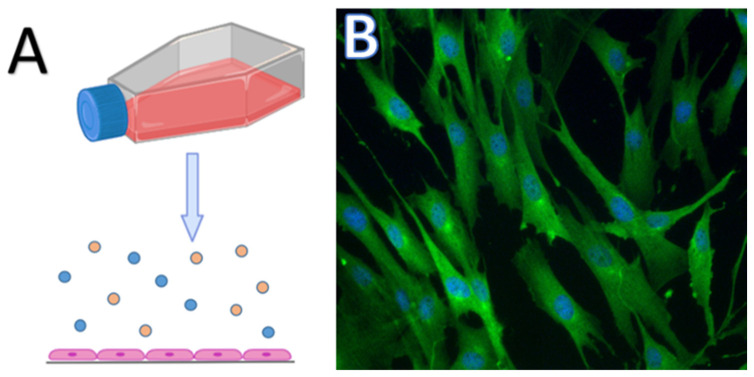
Representative images of CMT primary 2D culture. (**A**)—Schematic representation of adherent cells growing in monolayer with an unlimited supply of nutrients and oxygen in culture flask. (**B**)—Result of immunocytochemical staining of CMT primary culture cells using DAPI staining solution (blue nuclei) and Anti-Mucin MoAb (green cytoplasm). Magnification: 400×.

**Figure 2 cells-13-00695-f002:**
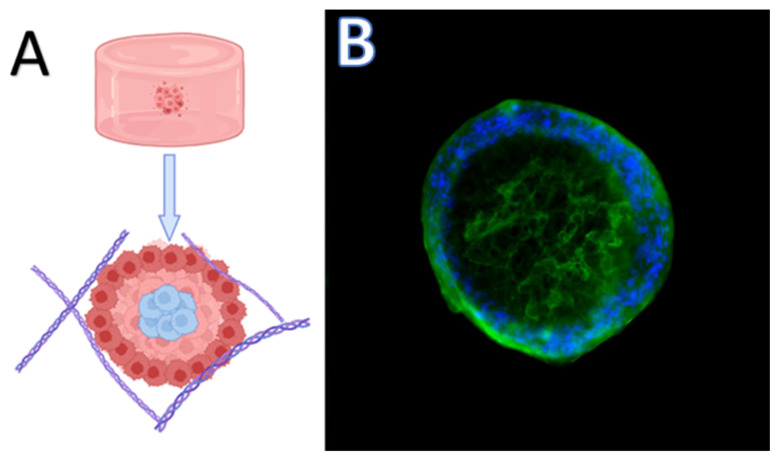
Representative images of a 3D CMT tumouroid. (**A**)—Schematic representation of 3D tumouroid with different access to oxygen and nutrients. Arrow pointing to the morphology of tumouroid. The outer layer with active proliferating cells (red cells), the middle layer of quiescent cells (pink cells) and a necrotic core (blue cells). (**B**)—Result of immunocytochemical staining of CMT tumouroid derived from primary culture using DAPI staining solution (blue nuclei) and Anti-Mucin MoAb (green cytoplasm). Magnification: 100×.

**Figure 3 cells-13-00695-f003:**
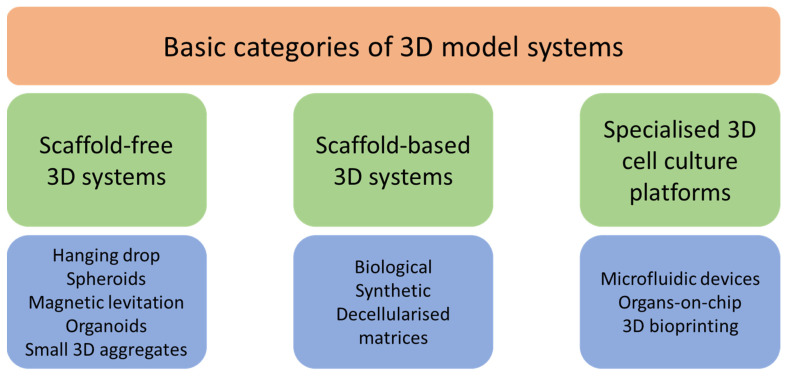
Different 3D cell culture systems [45].

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
