# Peer review of "Three-Dimensional Cultivation a Valuable Tool for Modelling Canine Mammary Gland Tumour Behaviour In Vitro"

_cells, 2024, doi:10.3390/cells13080695_

Round 1

Reviewer 1 Report

Comments and Suggestions for Authors

The authors reviewed 3D cell cultivation tools for modelling CMT behavior. However, this review is too simple and general, which does not adequately reflect the authors’ comprehensive understanding of this field. Therefore, it is not recommended for publication in Cells. Here are some comments:

1. It’s better to use the full name of CMT instead of the abbreviation in the title.

2. The section number and the title on page 3 line 99, page 4 line 134, page 7 line 294, and page 8 line 316 are confusing to the reader and should be corrected.

3. The title highlights the 3D cultivation tool for modelling CMT behavior. Thus, the authors should explain the requirement for the 3D cell culture environment for CMT modelling, which are significant in guiding the design of the 3D culture system. The subsequent sections for each specific tool should be organized closely to those requirements rather than general description of their properties.

4. At the end of section 4.2.1, the authors should comment on how to select the hydrogel material to construct the 3D system based on the requirement of research purposes.

5. Sections 4.3.1 and 4.3.2 are too simple and general. The authors should inculde more details about these tools.

Author Response

We appreciate all of your insightful and beneficial feedback on our paper. All the
recommendations have been consdered and manuscript revised. All included changes are highlighted yellow.
Please, find our point-by-point responses bellow.

Reviewer 2 Report

Comments and Suggestions for Authors

In this manuscript, the authors reviewed the recent advancement in the 3D cultivation technologies for modelling the behaviour of mammary gland tumours. The article is well organized and written. Before considering this manuscript for publication, the authors should consider the following points in any revision as follows:

1.      Some abbreviations should be defined when they first appear in the maintext, such as “CMT”, etc.

2.      In the second or third level title, there is an additional dot in the middle of 2D or 3D.

3.      The author should add more relevant figures when describing the relevant content to provide readers with a more intuitive presentation. For example, when describing the 3D cultivation using Scaffold-free 3D systems, Scaffold-based 3D systems, and Specialized 3D cell culture platforms.

4.      Some recent publications related to the 3D cultivation should be cited, such as Exploration 2023, 3, 20210043; Exploration 2021, 1, 20210036; Chinese Chemical Letters 2023, 34, 107573, etc.

Author Response

We appreciate all your helpful criticism on our paper. The manuscript has been
rewritten after considering all the recommendations. Every change that was included is
indicated in yellow.
Please, find our point-by-point responses bellow.

Round 2

Reviewer 1 Report

Comments and Suggestions for Authors

The authors have addressed most of my comments.

Reviewer 2 Report

Comments and Suggestions for Authors

The authors have revised the manuscript carefully according to the comments of the reviewers, and the quality of the manuscript has been improved. I suggest the acceptance of the manuscript.